# Study of Intracellular Peptides of the Central Nervous System of Zebrafish (*Danio rerio*) in a Parkinson’s Disease Model

**DOI:** 10.3390/ijms26052017

**Published:** 2025-02-26

**Authors:** Louise O. Fiametti, Camilla A. Franco, Leticia O. C. Nunes, Leandro M. de Castro, Norival A. Santos-Filho

**Affiliations:** 1Institute of Chemistry, São Paulo State University, Araraquara 14800-060, Brazil; louise.fiametti@unesp.br (L.O.F.); camilla.franco@unesp.br (C.A.F.); leticia.catarin@unesp.br (L.O.C.N.); 2School of Pharmaceutical Sciences, São Paulo State University, Araraquara 14800-903, Brazil; 3Institute of Biosciences, São Paulo State University, São Vicente 11380-972, Brazil; leandro.mantovani@unesp.br

**Keywords:** Parkinson’s disease, intracellular peptides, zebrafish larvae

## Abstract

Although peptides have been shown to have biological functions in neurodegenerative diseases, their role in Parkinson’s disease has been understudied. A previous study by our group, which used a 6-hydroxydopamine zebrafish model, suggested that nine intracellular peptides may play a part in this condition. In this context, our aim is to better understand the role of five of these nine peptides. The selection of peptides was made based on their precursor proteins, which are fatty acid binding protein 7, mitochondrial ribosomal protein S36, MARCKS-related protein 1-B, excitatory amino acid transporter 2 and thymosin beta-4. The peptides were chemically synthesized in solid phase and characterized by high-performance liquid chromatography and mass spectrometry. Circular dichroism was performed to determine the secondary structure of each peptide, which showed that all five peptides maintain a random structure in the aqueous solutions that were studied. Two molecules show a helical profile in trifluoroethanol, a known structuring agent. Cell viability by the MTT assay indicates that all five peptides are not cytotoxic in all concentrations tested in both mouse and human cell lines. Behavioral assay using a 6-OHDA zebrafish larvae model suggest that all peptides help in the recovery of motor function with 24 h treatment at two concentrations. Three peptides showed a complete recovery from the 6-OHDA-induced motor impairment. Further studies are needed to better understand the mechanism of action of these peptides and whether they are truly a potential ally against Parkinson’s disease.

## 1. Introduction

Parkinson’s disease (PD) is the second most common neurodegenerative disease in the world [1]. Its cause remains unknown and there is no cure [2]. Treatment options include a variety of pharmacological interventions, but patient adherence to medication is usually low due to a variety of side effects including nausea, orthostatic hypotension and narcolepsy, among others [3]. Levodopa, a drug introduced into the market in 1960, still provides the most significant improvement in motor symptoms, but its long-term use can worsen involuntary and uncontrolled muscle movements (dyskinesia) and motor fluctuations [4]. Thus, it is important to better understand PD and continue the research for new treatments.

A model that has been widely used in PD research is 6-hydroxydopamine (6-OHDA), a neurotoxin that is structurally similar to dopamine but is toxic to dopaminergic neurons, causing oxidative stress and neuroinflammation [5,6]. It is possible to treat animals with 6-OHDA, such as rodents [7,8,9] and fish, including zebrafish [10,11,12].

The use of zebrafish has become an alternative for the development of new drugs thanks to the improvement and standardization of the model in studies involving various behavioral repertoires in the field of neuropsychiatry [13]. For neuroscience in particular, the use of zebrafish as a model has allowed the correlation of physiological processes and behavioral responses with underlying molecular pathways or signaling cascades. This allows for a better understanding of the homeostasis mechanisms of the central nervous system, as well as pathological processes, such as in the case of neurodegenerative diseases [14].

In this context, a previous study by Fiametti et al. [10] used peptidomics to compare the peptide profile of 6-OHDA or saline-treated adult zebrafish brain. They showed that nine out of one hundred eighteen intracellular peptides identified in zebrafish brain were significantly altered in the 6-OHDA group. This suggests that these peptides may be involved in neurodegenerative processes, but further studies are needed to better understand if and how this involvement actually occurs. Five of the nine peptides mentioned were chosen for further studies. The peptides were selected based on their precursor proteins.

The first peptide chosen was GVGFATRQV (p-FABP7), which derived from the fatty acid binding protein 7 brain a (FABP7) [10]. According to Islam et al. [15], increased FABP7 elevates the expression of antioxidant enzymes and protects against apoptosis signaling. Ebrahimi et al. [16] showed that FABP7 deficiency causes abnormalities in dendritic neuronal morphology and affects excitatory synaptic functions. Islam et al. [15] also showed that another protein of the same family, fatty acid binding protein 3 (FABP3), is critical for the uptake of alpha-synuclein into dopaminergic neurons, a molecule found in Lewy bodies [2].

The second sequence that was selected was TLDNEMDYIQRGGPE (p-MRPS36), derived from mitochondrial ribosomal protein S36 (MRPS36) [10]. A study by Hevler et al. [17] suggests that MRPS36 evolved as an E3 adaptor protein, functionally replacing the peripheral subunit binding domain in the eukaryotic 2-oxoglutarate dehydrogenase complex. In other words, MRPS36 may have a role in the generation of NADH in aerobic respiration. Given the correlation between neurodegenerative diseases and reduced mitochondrial activity [18], MRPS36 can possibly have a role in PD.

The third peptide, TKQTEETNSTPAPSEQKE (p-MARCKS1B), is derived from the C-terminal region of the myristoylated alanine-rich C kinase substrate protein 1-B (MARCKS-related protein 1-B) [10]. The proteins of this family control cell movement by regulating the actin cytoskeleton, which is involved in the formation of filopodia and lamellipodia, and are highly conserved among vertebrates. MARCKS-related proteins play various roles related to brain growth, neuronal migration, neurite outgrowth, neurotransmitter release and synaptic plasticity [19].

The fourth sequence chosen, HESHLEPIE (p-EEAT2), is derived from the excitatory amino acid transporter 2 (EEAT2) [10], a protein with an essential role in the recycling of glutamate, the most abundant neurotransmitter in the mammalian brain [20]. A study by Trudler et al. [21] suggests that alpha-synuclein oligomers present in Lewy bodies induce glutamate release from astrocytes, leading to excessive extrasynaptic N-methyl-D-aspartate receptors, which are associated with neurodegeneration. Thus, glutamate recycling has an important role in controlling a neurodegenerative scenario.

Lastly, TIEQEKQAGSS (p-Tbeta4) is derived from thymosin beta-4 (Tbeta4) [10], a 43 amino acid peptide whose main function is to regulate actin polymerization, which is necessary for organogenesis and cell motility [22]. In addition, studies with Tbeta4 suggest that it is a peptide that supports myocardial recovery in post-infarction cases through cell migration [23,24], as well as in corneal [25] and liver recovery [26]. There is also evidence that Tbeta4 helps in the treatment of multiple sclerosis by restoring neuronal functionality [27], as well as in the improvement of injuries induced by oxygen-glucose deprivation [28] and reversing cognitive impairment in rats [29]. In other words, Tbeta4 may be an ally in neuronal recovery, and this study looks towards understanding if this derived peptide is also an ally.

As with other therapeutic strategies being studied, peptides may present some concerns such as stability, toxicity and immunogenicity [30]. However, they have some advantages over other small molecules, such as high specificity, high biological activities and high membrane penetration ability [31]. Such advantages result in about 100 peptides approved by the Food and Drug Administration (FDA) for various applications [32]. Thus, the hypothesis that one or more peptides altered in the study by Fiametti et al. [10] may have a beneficial effect in PD treatment is worthy of further investigation.

In order to perform further studies, all five peptides were synthesized, purified and characterized as described below. Cell viability by the MTT assays and behavioral assays in a 6-OHDA zebrafish larvae model were carried out to better understand the potential action of these peptides.

## 2. Results and Discussion

### 2.1. Synthesis, Purification and Characterization of Peptides

Peptides used in this study are the result of degradation by protein metabolism [10], which results in an acid C-terminal. For this reason, Wang resins were chosen for synthesis since, after cleavage in acid medium, their final peptide results in a COOH C-terminal. All peptides were synthesized, purified to >95% purity, and successfully characterized, as shown in Figure 1.

### 2.2. Mass Spectrometry

Mass spectrometry was used to confirm that the synthesized and purified material was the material of interest (Figure 2). To perform this, the mass/charge ratio of each peptide was calculated and compared to the value of each peak shown in the spectrum obtained. If the numbers for the mass/charge ratio and at least one peak in the spectrum are the same, this indicates that the molecular weight of the material is the material of interest, or in other words, that the synthesis was performed correctly.

### 2.3. Circular Dichroism

A circular dichroism (CD) spectrum indicates an all-helix structure when there are two negative bands at ~222 and ~208 nm, plus a positive band at ~190 nm. An all-*β*-sheet molecule spectrum shows a negative band between 210 and 220 nm and a positive band between 195 and 200 nm. A random-structured molecule has a negative band at 200 nm [33]. In this study, CD was performed at three different pHs (5, 7.4 and 10) in order to observe possible structure modification and peptide stability and also in trifluoroethanol (TFE), a solvent that is known to be structuring and stabilizing [34,35] (Figure 3).

FABP7 has a *β*-sheet folding for most of its structure but presents two close regions in helix, one of which happens to be where p-FABP7 is located [36]. However, the CD spectrum of p-FABP7 (Figure 3A) indicates a random structure for this peptide in all pHs studied and also in TFE.

MRPS36 has a random structure with four distant regions that evolve into an *α*-helix [17]. The CD spectrum of p-MRPS36 (Figure 3B) shows that this peptide remains with a random structure in all pHs studied, but in TFE the profile changes to a helix profile, showing that this configuration is possible.

The CD spectra for p-MARCKS1B (Figure 3C) and p-EEAT2 (Figure 3D) are similar: both show that in the three pHs studied and also in TFE both peptides remain in a random form. MARCKS-related protein 1B has a mostly random structure [37], while EEAT2 has a mostly *α*-helix structure [38].

Regarding Tbeta4, the precursor 43-amino acid peptide has a mostly random form but has two helixes throughout its structures [39]. The CD of p-Tbeta4 (Figure 3E) indicates that the derived peptide maintains a random structure at the pHs studied, but in TFE the profile changes to an *α*-helix.

### 2.4. Cell Viability by MTT Assay

Neuro 2A is a mouse neuroblast cell line [40] widely used in neuroscience research, while SHSY5Y is a cloned subline of human neuroblastoma [41] and is the most commonly used cell line in in vitro research about PD [42]. To investigate whether the peptides are cytotoxic, the MTT cytotoxicity assay was performed on both cell lines (Figure 4 and Figure 5). Peptide concentrations started at 512 µM and followed a serial dilution down to 1 µM. DMSO 10% was used as positive control.

Treatment in Neuro 2A cells (Figure 4) indicates that all peptides do not present cytotoxicity. This includes all concentrations tested, even the highest at 512 µM. Figure 5 shows that the peptides did not show cytotoxicity in the human neuroblastoma cell line either.

### 2.5. Behavioral Assay

It is well described in the literature that neurotoxicity of 6-OHDA causes behavioral changes in animals such as rodents [7,8,9] and fish [11,12,43,44]. The main parameters observed in these studies are mean speed and distance traveled during a specific time period. This can be a strategy to either verify the efficacy of treatment with 6-OHDA or to observe the action of different treatments and if they have an effect on behavior.

As shown in Figure 6 and Figure 7, treatment with all five peptides showed a significant improvement in mean speed (mm/s) and distance traveled (mm) compared to the 6-OHDA group at the two highest concentrations studied, 40 µM and 50 µM. Sequences p-FABP7, p-MPRS36 and p-Tbeta4 showed no significant difference compared to the control group at the highest concentration studied.

This suggests that the peptides help recover from the movement impairment caused by 6-OHDA. The results of the behavioral assay reinforce the lack of cytotoxicity of these peptides given that all groups observed had their physical capabilities improved, especially at higher concentrations. Three peptides, p-FABP7, p-MRPS36 and p-Tbeta4, showed statistically equal results to the control group in both parameters observed when treated with 50 µM, suggesting a complete recovery of mean speed and total distance swum. The precursor proteins of these three peptides have been shown to be involved in neurological functions in different scenarios, as mentioned above.

Further studies are needed to better understand the mechanism of action of these five peptides and whether or not these results are related to the role of each precursor protein. However, these initial data suggest that they are safe and effective molecules against 6-OHDA-Parkinson’s-like motor symptoms.

## 3. Materials and Methods

### 3.1. Peptide Synthesis

Synthesis was performed by solid-phase peptide synthesis [45] following the fluorenylmethyloxycarboxyl (Fmoc) chemistry protocol. Wang resins previously coupled to the first amino acid were used for all five peptides (Table 1). Coupling was made with N,N-diisopropylcarbodiimilde (DIC) and N-hydroxybenzotriazole (HOBt).

### 3.2. Purification and Characterization

The synthesized peptides were purified by high-performance liquid chromatography (HPLC) on a Shimadzu chromatograph in semi-preparative mode on a C18 reversed-phase column (Phenomenex 1 × 25 cm) at a flow rate of 5 mL/min in different methods (Table 2). The purity degree of the fractions was determined on a Shimadzu chromatograph on an analytical column (0.4 × 25 cm) of C18 reversed phase (Kromassil) in a gradient of 5–95% of solution B in 30 min at a flow rate of 1 mL/min. Solutions used for purification and verification of the peptide purity were 0.045% trifluoroacetic acid (TFA) in ultrapure water (solution A) and 0.036% TFA in acetonitrile (solution B).

### 3.3. Mass Spectrometry

To confirm the correct synthesis of peptides, mass spectra were obtained on an LCQ FLEET ThermoScientific mass spectrophotometer, with direct injection electrospray ionization, in positive detection mode.

### 3.4. Circular Dichroism

Circular dichroism (CD) spectroscopy was performed on a Jasco J-715 spectropolarimeter. CD spectra were obtained in aqueous solutions at pHs 5 (sodium acetate buffer 0.1 mol/L), 7.4 (PBS, phosphate buffer saline 0.027 mol/L) and 10 (borax buffer 0.1 mol/L) and in trifluoroethanol (TFE), a known structuring solvent [34,35].

### 3.5. Cell Viability by the MTT Assay

Neuro 2A and SHSY5Y cells were maintained in an incubator at 37 °C (atmosphere containing 95% air and 5% CO_2_) in DMEM medium (Dulbecco’s modified eagle medium, Gibco^®^, Grand Island, NY, USA) and DMEM/F12 medium (1:1 mixture of DMEM and Ham’s F12 media, Gibco^®^, Grand Island, NY, USA), respectively, plus 100.000 units/L of penicillin, 0.1 g/L of streptomycin and 10% fetal bovine serum (FBS, Gibco^®^, Grand Island, NY, USA). After reaching ~80% confluency, cells were transferred to a 96-well plate, at 0.01 × 10^6^ cell density. When ~80% confluency/well was reached, the medium was changed to DMEM or DMEM/F12 without antibiotics and FBS, plus each peptide in different concentrations. Dimethyl sulfoxide (DMSO) 10% was used as the positive control. After 24 h, the medium was removed, and each well was carefully washed with PBS. Then 5 mg/mL MTT (3-(4,5-dimethylthiazol-2-yl)-2-,5-diphenyltetrazolium bromide, Thermo Fisher Scientific, Waltham, MA, USA) in DMEM or DMEM/F12 was added to each well, and cells were incubated at 37 °C for 1 h. The medium was removed once more and substituted by DMSO. Plates were agitated on a shaker for approximately 15 min. Absorbance reading was performed at 570 nm on a VERSAMax spectrophotometer (Molecular Devices, LLC, San Jose, CA, USA). Statistical analysis was performed using a one-way analysis of variance (ANOVA) test followed by Dunnett’s test for multiple comparisons. Data were statistically analyzed using GraphPad Prism Software Version 8.0.2 (GraphPad Software Inc., San Diego, CA, USA).

### 3.6. Animals

Wild-type adult *D. rerio* (zebrafish) were obtained from a licensed commercial distributor. They were maintained on a 14 h light–10 h dark photoperiod in fresh filtered water at 28 °C and fed twice a day with dry food and once a day with *Artemia* sp. Reproduction was performed in individual fish tanks in a 1 male–1 female proportion, with pebbles and artificial plants. Eggs were collected, selected and maintained in an incubator at 28 °C until 4 days post fertilization (dpf). This study followed the guidelines of the National Council for Animal Experimentation Control (CONCEA) and was approved by the Ethics Commission for Animal Use (CEUA) at the Bioscience Institute of São Paulo State University (São Vicente, Brazil: Protocol Number 06/2023, approved on 22 June 2023).

### 3.7. Behavioral Assay

Zebrafish (*Danio rerio*) larvae at 4 dpf were transferred to a 96-well plate, a single larva per well. Larvae were treated with 750 µM 6-OHDA in system water for 24 h [46] in an incubator at 28 °C. Then, solution was changed to system water with each peptide in different concentrations—10 µM, 20 µM, 30 µM, 40 µM and 50 µM [47]—and larvae were incubated at 28 °C for another 24 h. Finally, each larva was filmed for 5 min, and videos were processed by an automated tracking system (ToxTrac Version 2.96). Analysis parameters were mean speed (mm/s) and total distance swum (mm) (Figure 8) [45]. Statistical analysis was performed using a one-way analysis of variance (ANOVA) test followed by Dunnett’s test for multiple comparisons. Data were statistically analyzed using GraphPad Prism Software (GraphPad Software Inc., San Diego, CA, USA).

## 4. Conclusions

All five peptides were successfully synthesized, purified and characterized. In aqueous solutions at pHs 5, 7.4 and 10, all peptides maintained a random secondary structure. In TFE, p-MRPS36 and p-Tbeta4 presented an *α*-helix profile. None of the peptides presented cytotoxicity in either cell line in all concentrations studied, which suggests that they can be used safely. Regarding zebrafish behavior, the performed assay indicates that all peptides help recover physical impairment caused by 6-OHDA in the two highest concentrations observed (40 µM and 50 µM). Three peptides showed no statistical difference when comparing treatment with 50 µM to the control group. This can possibly indicate that p-FABP7, p-MRPS36 and p-Tbeta4 led to a complete recovery of the physical impairment caused by 6-OHDA. Further studies are needed to better understand the mechanisms of these peptides and also to ensure the safety of these molecules for long-term treatment.

## Figures and Tables

**Figure 1 ijms-26-02017-f001:**
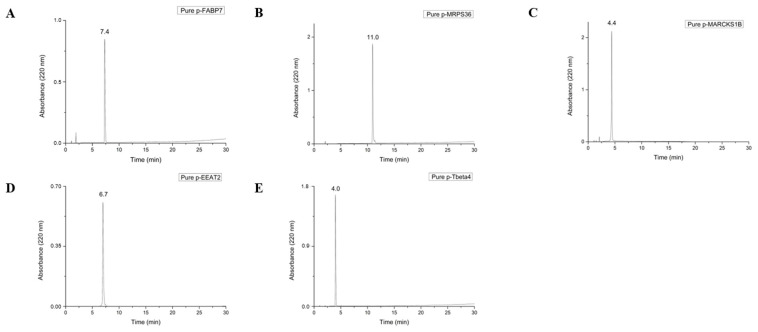
Chromatograms of pure peptides p-FABP7 (**A**), p-MRPS36 (**B**), p-MARCKS1B (**C**), p-EEAT2 (**D**) and p-Tbeta4 (**E**). The degree of purity of the fractions was determined on a Shimadzu chromatograph, using a 0.46 × 25 cm C18 reversed-phase analytical column (Kromasil). Solutions used for purification and verification of the purity degree of the peptides were 0.045% TFA in ultrapure water (solution A) and 0.036% TFA in acetonitrile (solution B) in a 30 min run at a flow rate of 1 mL/min in a 5–95% of solution B gradient.

**Figure 2 ijms-26-02017-f002:**
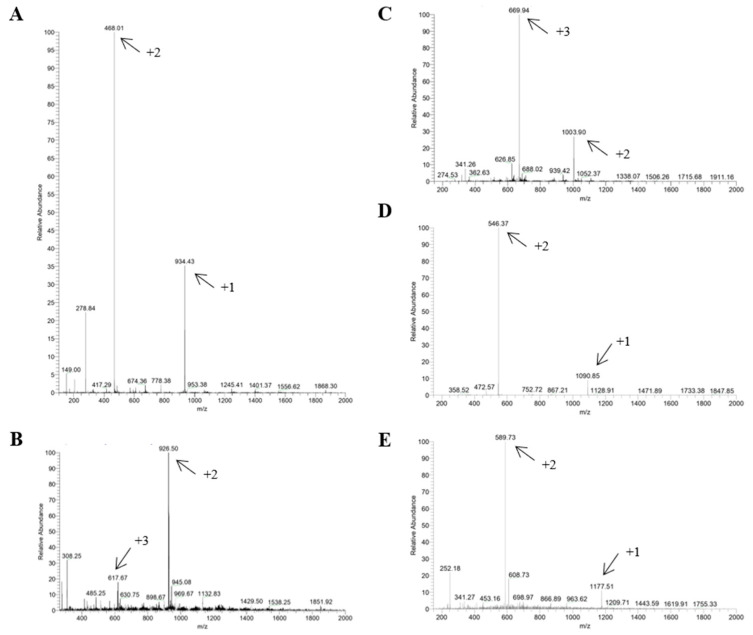
Mass spectra of p-FABP7 (**A**), p-MRPS36 (**B**), p-MARCKS1B (**C**), p-EEAT (**D**) and p-Tbeta4 (**E**). Arrows indicate peaks that confirm correct synthesis according to mass/charge ratio. Mass spectra were obtained by direct injection electrospray ionization in positive detection mode.

**Figure 3 ijms-26-02017-f003:**
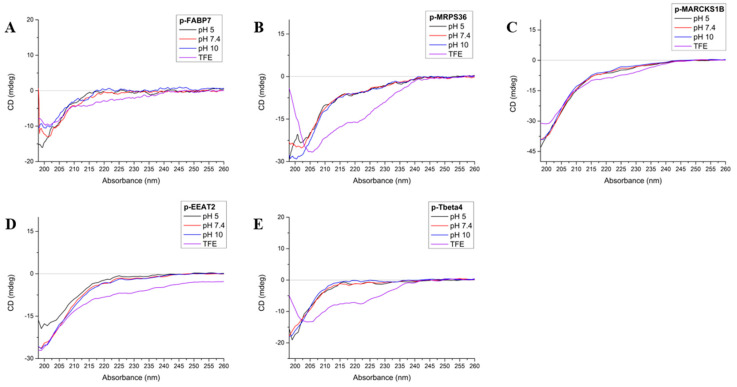
Circular dichroism spectra of p-FABP7 (**A**), p_MRSP36 (**B**), p-MARCKS1B (**C**), p-EEAT2 (**D**) and p-Tbeta4 (**E**) at pHs 5 (black), 7.4 (red), 10 (blue) and in TFE (violet).

**Figure 4 ijms-26-02017-f004:**
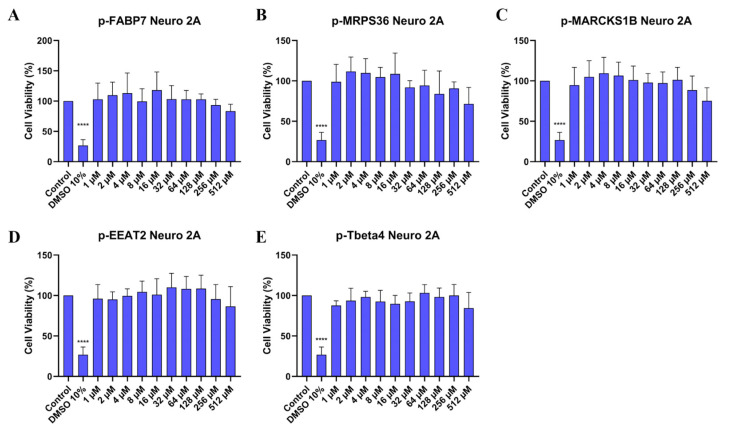
Cell viability (%) of Neuro 2A cells treated in different concentrations of p-FABP7 (**A**), p-MRPS36 (**B**), p-MARCKS1B (**C**), p-EAAT2 (**D**) and p-Tbeta4 (**E**). Experiments were performed in duplicate on three different occasions. ****, *p* < 0.0001.

**Figure 5 ijms-26-02017-f005:**
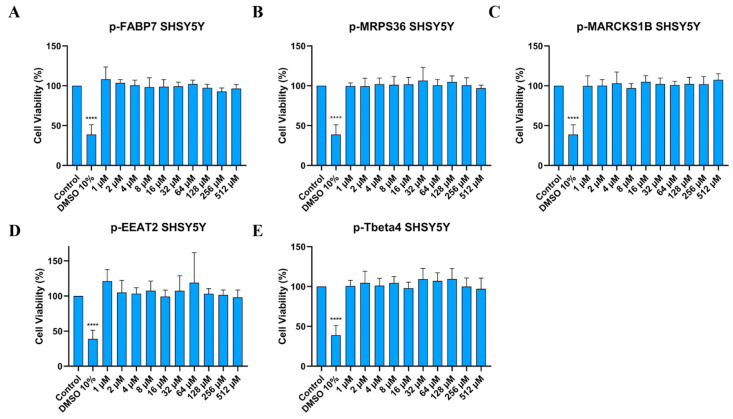
Cell viability (%) of SHSY5Y cells treated in different concentrations of p-FABP7 (**A**), p-MRPS36 (**B**), p-MARCKS1B (**C**), p-EAAT2 (**D**) and p-Tbeta4 (**E**). Experiments were performed in duplicate on three different occasions. ****, *p* < 0.0001.

**Figure 6 ijms-26-02017-f006:**
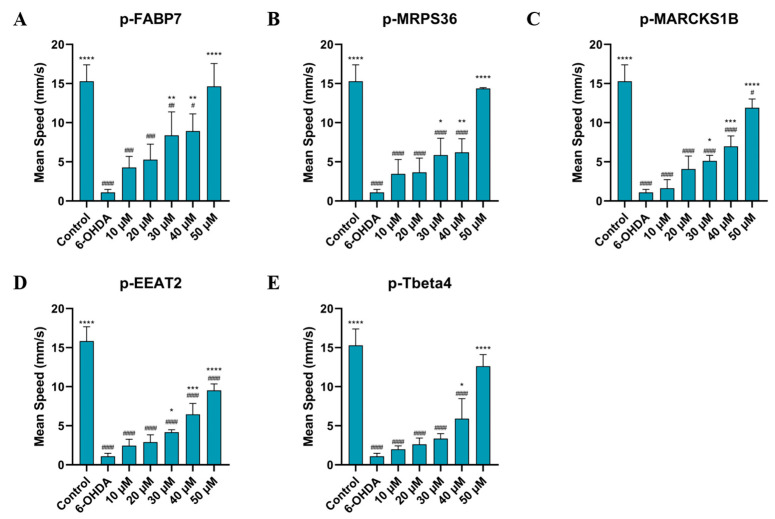
Mean speed of zebrafish treated with 750 µM for 24 h, then by p-FABP7 (**A**), p-MRPS36 (**B**), p-MARCKS1B (**C**), p-EEAT2 (**D**) and p-Tbeta4 (**E**) at concentrations of 10 µM, 20 µM, 30 µM, 40 µM and 50 µM for 24 h. Experiments were performed with 4 embryos per group on three different occasions, adding up to 12 embryos per group. #, *p* < 0.01, ##, *p* < 0.001, ###, *p* < 0001, ####, *p* < 0.00001 compared to the control group. *, *p* < 0.01; **, *p* < 0.001; ***, *p* < 0.0001; ****, *p* < 0.00001 compared to the 6-OHDA group.

**Figure 7 ijms-26-02017-f007:**
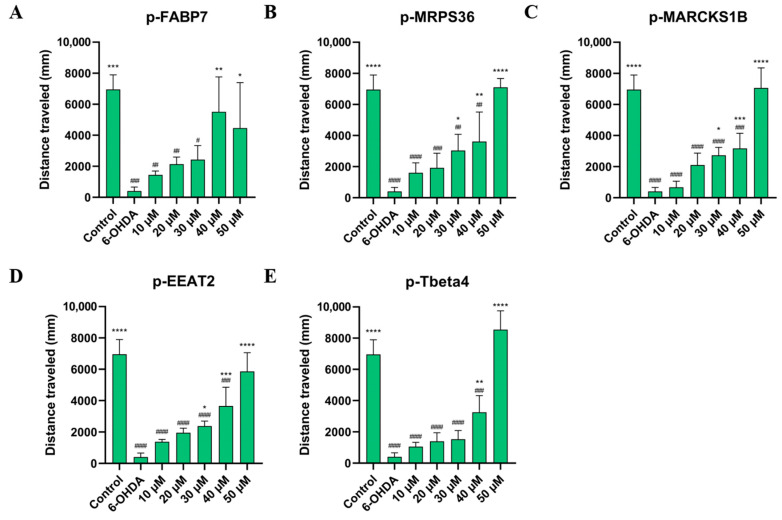
Distance traveled by zebrafish treated with 750 µM for 24 h, then by p-FABP7 (**A**), p-MRPS36 (**B**), p-MARCKS1B (**C**), p-EEAT2 (**D**) and p-Tbeta4 (**E**) at concentrations of 10 µM, 20 µM, 30 µM, 40 µM and 50 µM for 24 h. Experiments were performed with 4 embryos per group on three different occasions, adding up to 12 embryos per group. #, *p* < 0.01; ##, *p* < 0.001; ###, *p* < 0001; ####, *p* < 0.00001 compared to the control group. *, *p* < 0.01; **, *p* < 0.001; ***, *p* < 0.0001; ****, *p* < 0.00001 compared to the 6-OHDA group.

**Figure 8 ijms-26-02017-f008:**
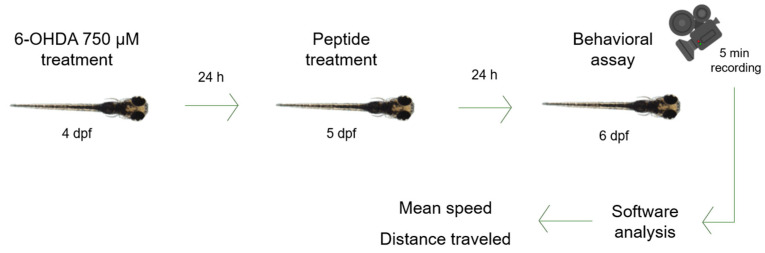
Zebrafish larva model. Zebrafish larvae were immersed in 6-OHDA 750 µM for 24 h, then in each peptide at 10 µM, 20 µM, 30 µM, 40 µM or 50 µM for 24 h. Videos were recorded for 5 min and then analyzed by software (ToxTrac Version 2.96). Parameters were mean speed and distance traveled.

**Table 1 ijms-26-02017-t001:** Resins used for synthesis of each peptide.

Peptide	Sequence	Precursor Protein	Resin
p-FABP7	GVGFATRQV	Fatty acid binding protein 7	Fmoc-Val-Wang
p-MRPS36	TLDLNEMDYIQRGGPE	Mitochondrial ribosomal protein S36	Fmoc-Glu(Otbu)-Wang
p-MARCKS1B	TKQTEETNSTPAPSEQKE	MARCKS-related protein 1B	Fmoc-Glu(Otbu)-Wang
p-EEAT2	HESHLEPIE	Excitatory amino acid transporter 2	Fmoc-Glu(Otbu)-Wang
p-Tbeta4	TIEQEKQAGSS	Thymosin beta-4	Fmoc-Ser(tBu)-Wang

**Table 2 ijms-26-02017-t002:** Purification protocol of each peptide.

Peptide	Retention Time (min)	Solution B (%)	Purification Method
p-FABP7	9.9	29.7	5% to 45% of Solution B in 120 min
p-MRPS36	11.0	33.0	5% to 55% of Solution B in 120 min
p-MARCKS1B	4.4	13.2	1% to 35% of Solution B in 90 min
p-EEAT2	6.7	20.1	1% to 35% of Solution B in 90 min
p-Tbeta4	4.0	12	1% to 25% of Solution B in 120 min

## Data Availability

Data supporting the conclusions of this study can be made available upon request from the corresponding author.

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
