# Peer review of "Study of Intracellular Peptides of the Central Nervous System of Zebrafish (Danio rerio) in a Parkinson’s Disease Model"

_ijms, 2025, doi:10.3390/ijms26052017_

Round 1
Reviewer 1 Report
Comments and Suggestions for Authors
In this study, five peptides were synthesized, purified and characterized as described below. Cell viability by the MTT assays and behavioral assay in a 6-OHDA zebrafish larvae model were carried out to understand the possible action of these peptides. However, some concerns should be given to show the significance of this review.
- In this study, behavioral assay with a 6-OHDA zebrafish larvae model suggest that all peptides help in the recovery of motor function with 24 h treatment at two concentrations. Three peptides showed a complete recovery from the physician impairment caused by 6-OHDA. The reason why three peptides showed a complete recovery from the physician impairment caused by 6-OHDA should be investigated in detail.
- The detailed mechanism that peptides help in the recovery of motor function should be detected at the gene expression level. Based on this gene expression level, the detailed signal pathway may participate in this process. It should be detected at the molecular level.
- Cell viability by the MTT Assay indicates that all five peptides are not cytotoxic in all concentrations studied, both in mouse and human cell lines. Why were the cell lines from zebrafish not used in this study, such as ZF4 cell line from zebrafish?
- The image of zebrafish larvae model should be added to show the effect of these peptides.
Minor revisions should be made on English.
Author Response
Comment 1: In this study, behavioral assay with a 6-OHDA zebrafish larvae model suggest that all peptides help in the recovery of motor function with 24 h treatment at two concentrations. Three peptides showed a complete recovery from the physician impairment caused by 6-OHDA. The reason why three peptides showed a complete recovery from the physician impairment caused by 6-OHDA should be investigated in detail.
Response 1: Thank you, the reviewer, for pointing this out. We agree with this comment, and this is our goal in the next steps of our research. The goal initially was to observe cytotoxicity and if the peptides would have an effect at all in order to continue the research with fewer peptides, thus reducing the number of animals used.
Comment 2: The detailed mechanism that peptides help in the recovery of motor function should be detected at the gene expression level. Based on this gene expression level, the detailed signal pathway may participate in this process. It should be detected at the molecular level.
Response 2: Thank you for your comment. This is already included in the next steps of our work. We intend to characterize the mechanism of action in a mouse model in the near future.
Comment 3: Cell viability by the MTT Assay indicates that all five peptides are not cytotoxic in all concentrations studied, both in mouse and human cell lines. Why were the cell lines from zebrafish not used in this study, such as ZF4 cell line from zebrafish?
Response 3: Thank you for this interesting question. Initially, mouse and human cell lines were chosen due to the fact that these are the animals we intend to continue the studies on. Furthermore, zebrafish larvae were treated with these peptides for 24 h in the behavioral assay, and all five of them helped improve the motor function of the larvae. This indicates that, at least in this acute treatment, they are not toxic to zebrafish.
Comment 4: The image of zebrafish larvae model should be added to show the effect of these peptides.
Response 4: Thank you for this suggestion. It has been provided on line 282, highlighted in yellow.

Reviewer 2 Report
Comments and Suggestions for Authors
The authors previously reported peptide profile of zebrafish brain in a 6-hydroxydopamine (6-OHDA)-induced Parkinson disease (PD) model. Role of peptides in PD brains has not been studies extensively yet. The authors previously found 9 intracellular peptides in the brain of zebrafish PD model. In the present study, the autors synthesized five of these 9 peptides: (1) fatty acid-binding protein 7 (p-FABP7); (2) mitochondrial ribosomal protein S36 (p-MRPS36); (3) MARCKS-related protein-1B (p-MARCKS1B); (4) excitatory amino acid transporter 2 (p-EEAT2); and (5) Thymosin beta-4 (p-Tbeta4). Cell viability by MTT Assay of all 5 peptides indicated no cytotoxic effects in mouce and human cell lines. Importantly, Behavioral Assay with 6-OHDA zebrafish larve PD model showed a complete recivery of motor function impairment caused by 6-OHDA.
All chemical studies are judged to be sound. Since the present reviewer is not a specialist in peptide synthesis, a reviewer of pepticde chemistry specialist would evaluate peptide chemistry in this study.
Since p-FABP7, p-MAPS36, and p-Tbeta4 led to a complete recovery of physical impairment of zebrafish caused by 6-OHDA, further studies in the PD models of mice and primates are highly promissing. For the possible clinical studies, the stabilty and safety for long term treatment, and route of administration should be precisely eaxamined for treating PD ptients. However, these peptides could be candidate drugs for prevention of progression and possible recovery in treating PD patients.
Author Response
Comment 1: All chemical studies are judged to be sound. Since the present reviewer is not a specialist in peptide synthesis, a reviewer of peptide chemistry specialist would evaluate peptide chemistry in this study.
Response 1: Thank you for your comment. Our lab is specialized in Solid Phase Peptide Synthesis and we agree that it is important.
Comment 2: Since p-FABP7, p-MAPS36, and p-Tbeta4 led to a complete recovery of physical impairment of zebrafish caused by 6-OHDA, further studies in the PD models of mice and primates are highly promising. For the possible clinical studies, the stability and safety for long term treatment, and route of administration should be precisely examined for treating PD patients. However, these peptides could be candidate drugs for prevention of progression and possible recovery in treating PD patients.
Response 2: Thank you for pointing this out. We understand the importance of studying stability of these molecules, as well as molecular pathways and mechanisms of action involved. These goals are already included in the next step of our work.
